# Manipulations in Democracy?

**DOI:** 10.3390/bs14040315

**Published:** 2024-04-11

**Authors:** Ruth Ben-Yashar

**Affiliations:** Department of Economics, Bar Ilan University, Ramat Gan 52900, Israel; benyasr@biu.ac.il

**Keywords:** simple majority rule, sincere voting, multiple private signals, state of nature

## Abstract

Democracy is upheld through the principle of majority rule. To validate the application of democracy, it is imperative to assess the sincerity of voter decisions. When voter sincerity is compromised, manipulation may occur, thereby undermining the legitimacy of democratic processes. This paper presents a general version of a symmetric dichotomous choice model. Using simple majority rule, we show that when a voter receives one or more private signals, sincere voting is an equilibrium behavior. A slight change to this basic model may create an incentive to vote insincerely. We show that even in a more restricted model where every voter receives only one private signal whose level of precision is the same for all the voters but depends on the state of nature, voters may have an incentive to vote insincerely.

## 1. Introduction

A committee or group is entrusted with the responsibility of determining which of two potential states of nature is the actual one. As an illustration, they may be tasked with discerning whether a candidate may be suitable or not, a defendant may be guilty or innocent, a medical treatment may be suitable or not, or a project may be good or bad, and, in each case, the decision is whether to support or oppose the proposal.

This paper adopts the methodology pioneered by [1]. Condorcet established a structured approach for effectively amalgamating all the votes of individuals involved in a particular collective decision [2]. The seminal Condorcet jury theorem, CJT [1], focuses on binary choices and delineates the circumstances in which majority rule serves as an effective means of consolidating the collective decisions of all participants. This theorem can be broadened to encompass more intricate scenarios, including decision-makers receiving signals of varying quality and engaging in multi-stage decision-making processes (See [3,4,5,6,7,8,9,10]. Ref. [11] generalize the non-asymptotic part of the theorem. For a discussion on how individuals may be influenced by others’ decisions, see [12,13,14,15,16]). Condorcet’s theoretical framework, though straightforward, lays the groundwork for contemporary economic modeling endeavors, which strive to identify the optimal decision-making approach—namely, one that facilitates efficient information aggregation. (Ref. [7] define the optimal decision rule, i.e., the decision rule that maximizes the expected utility, in the context of symmetric dichotomous choice. Ref. [17] define the optimal decision rule in a more general framework that allows for asymmetric dichotomous choice). All the aforementioned studies operate under the assumption that each individual makes decisions based solely on their own signal, i.e., decides sincerely.

We operate under the assumption of a symmetric model, wherein the two states of nature carry equal prior probabilities and entail symmetric costs (benefits) associated with making an incorrect (correct) collective decision. Furthermore, all probabilities remain independent of the specific state of nature. The voters share a unified objective of arriving at the accurate collective decision, with each voter receiving one or more private signals. These signals may pertain to various facets of a given scenario, such as the technical feasibility of a project, its market potential, or the reliability and commitment of a candidate. The committee endeavors to amalgamate these individual signals in an efficient manner, striving to employ the optimal decision-making framework that maximizes the likelihood of reaching the correct decision. Under this optimal rule, it is rational for each voter to cast their vote in accordance with their private signals.

Simple majority rule, which requires the backing of more than half of the voters, stands as the prevailing voting method in democratic systems and thus garners significant attention within voting theory. It proves apt for facilitating widespread democratic engagement in collective decision-making processes, particularly when all voters share a common goal. An integral characteristic of majority rule lies in its efficiency, defined as the likelihood of arriving at the correct collective decision, provided it exists.

However, simple majority rule is, in general, not the optimal rule. Under this suboptimal decision rule, a potential problem is that each voter will vote in a way that reflects not necessarily his own signals but rather his belief about how the group will vote, if he thinks that the group will reach a better decision. Such strategic voting, under which voters may have an incentive not to tell the truth, may increase expected utility beyond the level of sincere voting. Numerous studies have extended Condorcet’s methodology to incorporate strategic decision-making dynamics, such as [18,19,20,21,22]. These investigations reveal instances where sincere voting leads to inefficiency. To the best of our understanding, the present paper represents the pioneer effort to demonstrate that, also in the general model where a voter may receive more than one private signal, if the two states of nature are symmetric, then using simple majority rule guarantees that each voter votes according to his private signals, i.e., votes sincerely.

That is, a voter does not find it worthwhile to deviate from his private signals. On the other hand, a slight change to the symmetry of the basic model described above may create an incentive to vote insincerely. In certain cases, even when the voters share common preferences, a voter may deviate from his private signals. Accordingly, we reassess the validity of sincere voting by using a more restricted model where each voter receives only one private signal whose level of precision is the same for all the voters but depends on the state of nature (i.e., the signal’s precision is higher in one state of nature than in the other). To the best of our knowledge, this paper is the first to show that under these conditions voters may have an incentive to vote insincerely (i.e., strategically). 

## 2. The Model

Consider a committee $N=\left\{1,\ldots ,\;2n+1\right\}$ that chooses between the alternatives 1 and −1. The resulting two equiprobable states of nature ω therefore satisfy $\omega \in \left\{1,-1\right\}.\;$Consider, for example, a panel of 2n+1 judges that hear a defendant, where the outcome is to convict or acquit (this assumption guarantees an odd number; it is also an acceptable assumption under the majority rule in order for there to be a decision). In one scenario, denoted as state of nature 1, the defendant is found guilty, while in the alternative scenario, state of nature −1, the defendant is found innocent. We contemplate two potential outcomes stemming from the decision-making process: one being correct, which is preferable for all judges involved, and the other being incorrect, which is undesirable. As is typical in decision-making scenarios, the superior alternative remains unidentified. A judge *i* receives 2$k_{i}$ + 1 independent private signals about the true state of nature. A signal can be either 1 or −1. Let $s_{i}\in {⋃}_{k_{i}}{\left\{-1,1\right\}}^{{2k}_{i}+1}\;$be judge *i*’s signal profile, where ${2k}_{i}$ + 1 (the number of signals of judge *i*) follows distribution *f*. f is the probability that judge i receives ${2k}_{i}+1$ signals. Each judge receives a different number of signals. For instance, variations in the proficiency levels of individual judges, their degree of focus on the trial proceedings, and incidental factors may lead to discrepancies in the quantity of information each judge acquires regarding the prevailing circumstances. The allocation of informational signals to judges adheres to an exogenous distribution, denoted by *f*. Each judge, denoted by *i*, undergoes an independent random draw resulting in a realization, denoted by $k_{i}$. A large $k_{i}$ may signify, for instance, a judge possessing exceptional talent who meticulously attends to the trial proceedings.

Each signal’s precision is $q\in \left\{\frac{1}{2},1\right\}$. That is, if the state is 1, then *q* is the probability that the signal is 1 and 1 − *q* is the probability that the signal is −1; whereas, if the state is −1, then *q* is the probability that the signal is −1 and 1 − *q* is the probability that the signal is 1. Note that in a symmetric model, *q* is independent of the state of nature. The panel of judges have to decide on the guilt or innocence of the defendant; i.e., they aim to aggregate their signals. An important aspect of this process is its efficiency, i.e., the probability of reaching the correct collective decision. According to the standard assumptions: (i) the judges’ goal is to reach this correct decision; (ii) the judges engage in discussions regarding the case, serving as a part of their learning process before the final decision; (iii) the judges vote independently. The collective probability of reaching a correct decision is based on the judges’ signals’ precision.

## 3. The Optimal Decision Rule

The effectiveness of group decision-making processes can be assessed through various metrics, one of which is efficiency. Efficiency, as defined by [23], pertains to the accuracy of the decision-making process, specifically its probability of leading to the correct decision. This criterion of efficiency serves as the focal point of our analysis of group decision-making. Note that we can focus on the collective probability because we assume that the net benefit (i.e., the benefit from making a correct decision minus the cost of making an incorrect one) is the same in each state. This assumption leads to the conclusion that maximum collective probability corresponds to maximum expected benefit. Hence, the group aims to aggregate its voters’ signals in an efficient manner, i.e., to use the optimal decision rule that maximizes the expected probability of reaching the correct decision. The optimal decision rule can be formulated as a function of the total number of private signals that the voters receive and the total number of 1-type signals among them; i.e., the final decision is 1 (−1) if the number of signals 1 (−1) of all the voters is more than half of all the signals. 

Under this rule, it is rational for the voters to send their private signals; see [24]. In a special case of this model where each voter receives only one signal, this result was achieved by [18]. They showed that under the optimal decision rule, sincere voting is in fact a Nash equilibrium. This finding effectively addresses the critique of the assumption of sincere voting under the optimal decision rule.

## 4. The Result

In practice, simple majority rule is used in general. Under this rule, each voter votes by using the simple majority rule for 1 or −1 based on his private signals, and then the group uses simple majority rule to aggregate the different votes of the voters. This rule is in general not optimal. One potential problem is that each voter will vote in a way that reflects not necessarily his private signals but rather his belief about how the group will vote, if he thinks that the group will reach a better decision. For example, assume that there are three voters, where the first voter has seven signals and three of them are 1, the second voter has three signals and one of them is 1, and the third voter has three signals and all of them are 1. In this setting, the optimal final decision is 1 (seven of the 13 signals are 1). However, if the group uses the suboptimal simple majority rule, then each voter votes in a way that reflects his private signals: the first and second voters vote −1 and the third voter votes 1, and the final decision is −1. In this setting, one or more voters may prefer not to vote according to their private signals. Such strategic voting, under which voters may have an incentive not to tell the truth, may increase expected utility beyond the level of sincere voting. Our result shows that under a symmetric model, using simple majority rule guarantees that each voter votes according to his private signals, i.e., votes sincerely. To the best of our knowledge, this paper is the first to show this result in the general model where a voter may receive more than one private signal.

**Proposition 1.** 
*In a symmetric model with simple majority rule, sincere voting is an equilibrium behavior.*


**Proof.** Consider a voter *i* who receives ${2k}_{i}+1$ signals. Under sincere voting, the probability that his vote is correct (i.e., vote 1 if the state is 1 and vote −1 if the state is −1) is
$$Q_{i}=\sum_{l\geq k_{i}+1}^{\;}\left(\begin{array}{c} 2k_{i}+1\\ l \end{array}\right)q^{l}{\left(1-q\right)}^{2k_{i}+1-l}.$$

Note that the lower bound $l\geq k_{i}+1$ is determined by the fact that each voter votes based on his private signals, and, therefore, at least $k_{i}+1$ out of ${2k}_{i}+1$ signals must be 1 or −1 for the voter to vote 1 or −1, respectively. Thus, the ex-ante probability that an individual vote is correct is $Q=\sum_{k_{i}}f\left(k\right)Q_{i}$, where ${2k}_{i}+1$ follows distribution *f.* Let $l_{i}$ be the number of 1-type signals of voter *i*. Then, voting for 1 is optimal if and only if
$$\frac{Prob\left(i\;is\;pivotal\;\right|\;\omega =1)}{Prob\left(i\;is\;pivotal\;\right|\;\omega =-1)}=\frac{Q^{n}{\left(1-Q\right)}^{n}q^{l_{i}}{\left(1-q\right)}^{2k_{i}+1-l_{i}}}{Q^{n}{\left(1-Q\right)}^{n}q^{2k_{i}+1-l_{i}}{\left(1-q\right)}^{l_{i}}}={\left(\frac{q}{1-q}\right)}^{2l_{i}-2k_{i}-1}$$

Voting for 1 is optimal if and only if $2l_{i}-2k_{i}-1\geq 0$, that is, iff $l_{i}>k_{i}-\frac{1}{2}$. In other words, voting for 1 is optimal if the number of 1 signals is greater than the number of −1 signals. 

Note that being a pivotal voter means that the final decision will be different if the voter changes his vote. If the probability of a voter being pivotal under state of nature 1 is greater than or equal to the probability of the voter being pivotal under state of nature −1, then voting for 1 is optimal. Since q≥1/2, the above inequality holds if and only if $l_{i}\geq k_{i}$. This implies that sincere voting is optimal (for more explanation, see Appendix A). □

Sincere voting is an equilibrium behavior whenever there is complete symmetry between the two states of the world under the simple majority rule. This is mainly because, under the above-mentioned assumptions, the probability of each voter being pivotal is the same in both states. In situations where the probability of being pivotal differs between the two states, it might be rational for the pivotal voter to assume that the state with the higher probability of pivotality is the prevailing one, thereby influencing their voting behavior irrespective of their signal. However, such a scenario cannot occur within a symmetric setting under the simple majority rule.

## 5. The Model in the Case Where the Signal’s Precision Depends on the State of Nature

The possible dependence of the signal’s precision on the state of nature may also play a significant role in reaching the correct decision. In many decision-making contexts, the signal’s precision is higher in one state of nature than in the other. To illustrate this, let us consider the composition of judges in a court setting tasked with adjudicating whether a defendant is guilty or innocent. Here, we delineate two states of nature: one where the defendant is indeed guilty, and the other where the defendant is innocent. When assessing the precision of the information signal, judges with a background primarily in general law, devoid of specific experience as prosecutors or defense lawyers prior to their judicial appointment, rely on their broad legal training and expertise. Consequently, they assess a signal’s precision regarding the defendant’s guilt or innocence independently of the state of nature. On the other hand, there are judges who bring extensive experience from their prior roles as prosecutors or defense attorneys before assuming the bench. These judges possess skills and insights that may vary depending on the specific state of nature. These judges may have state-dependent skills. The former prosecutor (defense lawyer) has a better sense of a signal’s precision when the defendant is guilty than when the defendant is innocent (when the defendant is innocent than when the defendant is guilty). Accordingly, we refer to judges with former experience as prosecutors or defense lawyers as having a signal’s precision that is state-dependent. Specifically, in state of nature 1, i.e., when the defendant is guilty, the signal’s precision is $q_{1}$, and, in state of nature −1, i.e., when the defendant is innocent, the signal’s precision is $q_{2}.\;$That is, if the state is 1, then $q_{1}$ is the probability that the signal is 1 and 1 − $q_{1}$ is the probability that the signal is −1; whereas, if the state is −1, then $q_{2}\;$is the probability that the signal is −1 and 1 − $q_{2}$ is the probability that the signal is 1. These probabilities represent the the signal’s precision such that for each judge, $\frac{q_{1}+q_{2}}{2}>\frac{1}{2}$ (see [17]). For judges whose signal’s precision is independent of the state of nature, the signal’s precision is $q=q_{1}=q_{2}$ (Refs. [25,26] relax the symmetry assumption with respect to the states of nature and allow the decision-making skills of each voter depend on the state of nature. Ref. [17] derive the optimal group decision rule under such an asymmetric setting).

## 6. The Optimal Decision Rule in the Case Where the Signal’s Precision Depends on the State of Nature

Under the assumption that each voter receives only one private signal whose level of precision is the same for all the voters but depends on the state of nature, the optimal rule is proved by [17]. The optimal rule is a qualified majority rule, such that the group decision is 1 if and only if the net number of group members who vote 1 is greater than or equal to a quota. The quota is a function of a bias that is caused by the signal’s different levels of precision corresponding to the two possible states of nature. Formally, the optimal quota given by [17] can be expressed as follows (for more details on the optimal quota, see [17]): $$\frac{-\frac{n}{2}ln\frac{q_{1}(1-q_{1})}{q_{2}(1-q_{2})}}{\frac{1}{2}(ln\frac{q_{1}}{\left(1-q_{1}\right)}+ln\frac{q_{2}}{\left(1-q_{2}\right)})}$$

## 7. The Result in the Case Where the Signal’s Precision Depends on the State of Nature 

The following proposition shows that, although our model is more restricted such that each voter receives only one private signal whose level of precision is the same for all the voters but depends on the state of nature, the voters may have an incentive to vote insincerely. To the best of our knowledge, this paper is the first to show that under these conditions, voters may have an incentive to vote insincerely (i.e., strategically). 

**Proposition 2.** *In a model with simple majority rule where the signal’s precision depends on the state of nature and the two states of nature are symmetric in their prior probability and in the cost (benefit) of making the wrong (correct) collective decision, sincere voting may not be an equilibrium behavior*.

**Proof.** Assume that three voters each have two probabilities, *q*_1_ = 0.9 and *q*_2_ = 0.6. According to [17], the optimal quota $\frac{-\frac{n}{2}ln\frac{q_{1}(1-q_{1})}{q_{2}(1-q_{2})}}{\frac{1}{2}(ln\frac{q_{1}}{\left(1-q_{1}\right)}+ln\frac{q_{2}}{\left(1-q_{2}\right)})}$ is 1.13. In other words, the final decision is 1 only if all three voters support this alternative. However, the final decision is −1 if only two voters support alternative 1. This is because the net number of group members who vote 1 is one, which is less than 1.13, and hence the final decision is −1. Under simple majority rule, each voter is pivotal when only one of the other two voters supports 1. Being a “pivotal” voter means that the group’s decision will be different if the voter changes his vote and votes −1 instead of 1. The incentive to vote strategically arises because his vote matters only if his vote is pivotal. Thus, a voter may choose to ignore his 1-type signal and instead vote −1. Such strategic voting can increase the expected utility beyond the level of voting sincerely. □

The phenomenon of strategic voting arises from the discrepancy in the likelihood of each voter being pivotal across different states of nature. When the probability of being pivotal is greater in one state, such as state −1, compared to another state, such as state 1, it may be rational for the voter to presume that the first state obtains and to vote accordingly, regardless of the signal.

## 8. Conclusions

This article offers insights into the justification for democracy. While democracy operates on the basis of majority rule, its validation hinges on the integrity of voter decisions. Instances of manipulation among individuals compromise the rationale behind employing majority rule and, consequently, the legitimacy of democracy itself. Sincere voting, whereby each voter votes according to his private signals, is an equilibrium behavior in a democracy in which simple majority rule is used to aggregate the different votes. We show that this result is valid also in our general model where a voter may receive more than one private signal. A slight change to this basic model may create an incentive to vote insincerely. We show that even in a more restricted model where each voter receives only one private signal whose level of precision is the same for all the voters but depends on the state of nature, voters may have an incentive to vote insincerely (i.e., strategically). That is, they will vote in a way that reflects not necessarily their private signals but rather their individual beliefs about how the group will vote, if they think that the group will reach a better decision.

## Data Availability

Data are contained within the article.

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
