# Peer review of "Manipulations in Democracy?"

_behavsci, 2024, doi:10.3390/bs14040315_

Round 1

Reviewer 1 Report

Comments and Suggestions for Authors

The paper is interesting, but not always clear and should be made more precise. Here are my suggestions:

line 55: What do you mean by "$k_i$ follows distribution $f$" ?

line 78, 79: the optimal final decision [skip "decision" once]

section 4: Here the majority rule is used twice. First by each individual voter $i$ to vote 1 or -1 given his $2 k_i + 1$ signals, next to determine the group decision of all voters. I think this should be mentioned explicitly.

line 87, Proposition 1: what do you mean by a symmetric model? You give an explanation in lines 20, 21. Do I understand correctly that in terms of section 5 it means that $q = q_1 = q_2$ ? If so, it helps to point this out before  presenting Proposition 1.

line 89: Proof: Suppose voter $i$ receives $2 k_i + 1$ signals.

line 90: "the probability that his vote is correct". Do you mean that his vote is the same as the final decision?

line 91: I think that instead of $Q_k$ you should write $Q_i$ or $Q_{k_i}$.

line 95: What do you mean by f(k)? and where does the $k$ come from? What is the relation between $k$ and $k-i$, if any ? Should $Q$ be replaced by $Q_i$ ?

line 96: Should this line be replaced by: Given signals $s_1, \ldots, s_{2k_{i} + 1}, let $l$ be the number of 1-type signals of voter $i$ ? I think it is more precise to use $l_i$ instead of $l$.

lines 125-128: this text is exactly the same as text in Section 4.

line 132: again, what do you mean by a symmetric model ? In the counterexample $q_1$ is different from $q_2$.

Reviewer 2 Report

Comments and Suggestions for Authors

The contribution to existing literature is not clear. Which results are novel? Simple majority rule is well studied. It should be emphasazed what was not known in the literature.

The paper is not self-explanatory. What is a goal of a voter? Section 3 refers to some another paper, but optimal rule should be  defined in the paper itself. In the proof of proposition 1, why " If the probability of a voter being pivotal under state of nature 1 is 100 greater than or equal to the probability of the voter being pivotal under state of nature -1, 101 then voting for 1 is optimal"?  Again, what is optimal?

Round 2

Reviewer 1 Report

Comments and Suggestions for Authors

The second version of this paper is clearly better than the first one. However, some more improvements are required or at least useful.

line 101: Let s_i \in {-1, 1}^{2k_i + 1}, i.e., s_i is a sequence of 2k_i + 1 signals 1 or -1. I think the union sign should be removed.

line 143: where 2k_i + 1 follows distribution f. What do you mean by this?

line 232: Q = \sigma_{i = 1}^{2n+ 1} f(k) Q_{k_i}, where 2k_i + 1 follows distribution f. Question: what is f(k), and where does k come from? By the way, I would use Q_i instead of Q_{k_i}.

line 233: voting for i is optimal [i instead of 1]

Author Response

Please find 

Reviewer 2 Report

Comments and Suggestions for Authors

The paper was improved in the revision.

Unfortunately, I still fail to understand the main result. As ii is written in the paper:

"When the probability of being pivotal is greater in one state, such as state -1, compared to another state, such as state 1,  it may be rational for the voter to presume that the first state obtains and to vote accordingly, regardless of the signal".

This is an explanation of the main result. But why being a judge pivotal forces him to vote against his own signal? Because he does not believe himself that his signal is correct? How does it follow from the proof?  If he believes that the true state of the world is 1, why to vote -1, even if he is pivotal?

Author Response

Please find att

Round 3

Reviewer 2 Report

Comments and Suggestions for Authors

I am satisfied with the current version.

Author Response

.